evolution/biomaterials

microstructure, phenotype, predator–prey interaction, snail shell, defence mechanism

**Author for correspondence:**
H. Le Ferrand
e-mail: hortense@ntu.edu.sg

# Structure–behaviour correlations between two genetically closely related snail species

## H. Le Ferrand[1] and Y. Morii[2,3]

[1]School of Mechanical and Aerospace Engineering, School of Materials Science and Engineering, Nanyang Technological University, 50 Nanyang Avenue, 639798, Singapore, Singapore
[2]Phenix Group, School of Agriculture and Environment, Massey University, Private Bag 11-222, Palmerston North 4410, New Zealand
[3]Department of Forest Science, Research Faculty of Agriculture, Hokkaido University, Sapporo, Hokkaido 0608589, Japan

(iD) HLF, 0000-0003-3017-9403

Species, through their structure and composition, have evolved to respond to environmental constraints. Predator–prey interactions are among environmental pressures that can lead to speciation, but it remains unclear how this pressure can be related to the material structure and performance. Recently, two land snails, *Karaftohelix editha* and *Karaftohelix gainesi*, were found to exhibit divergent phenotypes and responses to predation despite sharing the same habitat and most of their genome. Indeed, under attack from a beetle, *K. editha* snails retract into their shell whereas *K. gainesi* snails swing their shell. In this paper, we looked at the microstructure, composition, morphology and mechanics of the shells of those two species and discuss potential relationships between material structure and the snail defence behaviour. The results of this study provide additional arguments for the role of predator–prey interactions on speciation, as well as an unusual approach for the design of biomimetic structures adapted to a particular function.

## 1. Introduction

There is a general consensus that biological composites have evolved optimal structures to withstand the environmental constraints they are submitted to. For instance, the porous hierarchical architecture with oriented fibres found in bamboo stems contributes to its flexibility, toughness and resistance to transverse forces [1]. In other *biota*, gradients in mineral concentration, crystallinity level and hardness generate tortuous crack deflections to sustain repeated shocks without catastrophic

**Figure 1.** (*a*) Map indicating where the two species of snails *Karaftohelix editha* and *Karaftohelix gainesi* were collected. (*b*) Contrasting defence behaviours of the two snails under an attack by a beetle (in red). The snail body is highlighted in brown, whereas the snail shell is highlighted in orange. The graphs below the pictures indicate the retraction of the snail body inside its shell (distance *r*) as a function of time *t* (left) and the angle of the snail shell with respect to the snail body (angle $\theta_{swing}$) as a function of the time (right) for both species. (*c*) Cartoon representing the geometry of a shell of diameter *d* and height *l*. (*d*) Diameter *d* as a function of the height *l* (black, *K. editha* and white, *K. gainesi*). (*e*) Density $\rho$ as a function of the spire index $s = d/l$.

failure [2]. Despite less unique mechanical properties, the shells of land snails are particularly adequate to investigate how environmental constraints affect their structure and composition. Indeed, due to animal low mobility and strict habitat dependences, the phenotypic traits of land snails, including shell shape, colour, banding patterns and chirality, evolve rapidly and show high phenotypic variation inter species [3–9].

The mechanism of these phenotypic divergences, and the emergence of new species, have been a major concern in evolutionary biology since Darwin's time. Resource competition is considered to be one of the major causes promoting diversification (i.e. ecological speciation) [10], but predator–prey interactions can be an alternative factor [11–13]. So far, it remains unclear how predation induces phenotypic diversification of prey and how it might lead to speciation [10,13,14]. Although nature's strategies to develop weapons have been extensively reviewed, no correlations have been drawn between their structure, their evolution and environmental constraints [15]. To the best of the authors' knowledge, there has been no systematic study of attack or defence behaviours and how they relate to material structure and properties.

Recently, prey divergence in *Karaftohelix* land snails (Camaenidae (Bradybaenidae), Pulmonata) distributed in northern Japan and far-eastern Russia was reported, as a result of adaptation to specialist predators, carabid beetles (Carabidae, Coleoptera) [16,17]. This divergence led to contrasting phenotypic traits of shell morphology and defence behaviour in the snails. Among nine *Karaftohelix* species, genetically related and sharing the same habitat, two alternative anti-predator behaviours, passive and active, were observed [16,18]. Seven snail species exhibited a passive defence mechanism by withdrawing their soft bodies into their shells, whereas the other two species displayed an unusual active defence strategy: the swinging of their shells against the carabid beetles (figure 1*a*,*b*) [16]. The

difference between passive and active strategies was found to be reflected in their shell morphology, suggesting a correlation between behaviour and shell architecture to optimize the preferred defence strategy [16]. Since no obvious difference was found in the habitat of those species, it is likely that the phenotypic and behavioural divergence were induced by predation pressure.

It is known that gastropod shells can exhibit specific traits relative to defence strategies, such as thickness [19], shape [20], spikes [21], colours [19], aperture shape [22] and even acoustic sounds [23]. In this study, by comparing the defence behaviours and the shell structures of two *Karaftohelix* species living in Hokkaido area in the northern part of Japan, we aim at deciphering correlations between microstructure, morphology, properties, performance and defence strategy. The two species selected, *Karaftohelix editha* and *Karaftohelix gainesi*, are ideal materials for this purpose as they are genetically close species, yet they exhibit passive and active defence, respectively [15,16]. Another advantage of using *K. editha* and *K. gainesi* as models is that their interactions with predators are quite simple. In Hokkaido, only two malacophagous carabid beetles are dominant, *Damaster blaptoides* and *Acoptolabrus gehinii* (Carabidae, Coleoptera), and are probably the largest predators of *K. editha* and *K. gainesi*. Indeed, although birds and rodents are generally recognized important predators of land snails, this has been found to be not applicable in this case in Hokkaido [17].

The development of such diverging defence behaviours, therefore, should have functional meanings and can by hypothesized to result from interactions with the environment, hence in this case, the predators. In this study, we first compare the macroscopic characteristics of the shells in terms of dimensions, weight and balance. Secondly, we look at the mechanical properties and composition at macroscopic and microscopic levels. In the third part of the paper, we summarize our results and discuss the potential correlations between microstructure and properties of the shells and the snails' behaviours. Those correlations are interesting not only from an ecological and evolutionary perspective, but also in view of biomimicry and materials design.

# 2. Material and methods

## 2.1. Species collection and response to predatory attack

In total, six shells from living adults *Karaftohelix editha* and 11 from living adults *Karaftohelix gainesi* were studied. The snails were collected in Bibai, Hokkaido, Japan (43.3285° N, 141.9688° E) in June 2010. The list of locations and collection dates are referenced in the electronic supplementary material, table S1. The snails were placed in boiled water for 1 min, and their soft bodies were removed from their shells using tweezers following the procedure developed by Fukuda *et al.* [24]. After that, all shells were kept in dry state. The experiments to observe the response of the snails to a predator were carried out previously and published in Morri *et al.* [16]. The movies used to observe the response of the snails to predatory attacks from beetles can be found at this link: https://www.nature.com/articles/srep35600#Sec14. The movies 2, 6 and 9 were analysed for the swinging behaviour whereas the movie 8 was used to look at the retraction of the snail into its shell. The analysis of those movies was done using iMovie (Apple Inc., USA) and Image J (NIH, USA).

## 2.2. Macroscopic characterization

The macroscopic density of the shells was determined by measuring their dry weight and calculating the overall volume. The dry weights were measured after overnight drying in a vacuum oven operating at 30°C (Binder VD53, Fischer Scientific Pte Ltd). The angles of the columella at rest ($\theta_{rest}$), in action ($\theta_{act}$), and during the swinging of the shell ($\theta_{swing}$) were measured from pictures taken with a standard camera. For $\theta_{act}$, the shells were positioned on a horizontal substrate in their naturally stable position. All angles and dimensions were measured from images using the software Image J (NIH, USA). The $p$-values were calculated using a Student's $t$-test in Excel. The macroscopic observations of the surface cracks and other damage were carried out using a microscope (OLYMPUS SZX16) equipped with a camera (Qimaging, MicroPublisher 5.0RTV) and the software Image-Pro insight.

## 2.3. Electron microscopy

Scanning electron microscopy was performed on fractured cross-sections from the shells. First, stripes were cut from the shells at similar locations in the two species, then cross-sections were revealed by

brittle fracture. Fracture debris were removed using ultrasonication in a water bath. The cross-sections were mounted on SEM stubs, sputtered with 10 nm of Pt, and observed using a field emission scanning electron microscope (FESEM 7600F, JOEL Asia, Japan). Elemental composition was obtained using energy-dispersive X-rays (EDX) at 5 keV acceleration voltage. EDX spectra were recorded for 60 s and the data obtained averaged over three neighbouring areas of $100 \times 100\,\mu m$.

## 2.4. Mechanical characterization

Nanoindentation was carried out on polished cross-sections of the shells. In summary, pieces of shells that were cut at similar locations as observed in FESEM were cold-mounted in epoxy (Specifix, Struers, Germany), and mirror-polished using SiC papers of decreasing roughness and finally colloidal diamond suspensions. Polishing debris was washed off using water in an ultrasonication bath until all debris was removed. The specimens were then indented in a Triboindenter (TI-950 Hysitron-Bruker, USA) using a Berkovitch diamond probe of $2\,\mu m$ diameter calibrated on fused quartz. The maximum load applied was $5000\,\mu N$. Each indentation consisted of 5 s loading, 2 s holding at maximum load and 5 s unloading. The elastic moduli were calculated using the Olivier–Pharr method.

# 3. Results and discussion

## 3.1. Macroscopic and phenotypic divergence

The two snail species studied share the same habitat, have closely related genomes [18], but exhibit two distinct macroscopic phenotypes (figure 1).

The two species of snails have evolved in the same ecosystem in the island of Hokkaido, Japan (figure 1a). In this environment, they are facing similar predatory attacks mostly from carabid beetles *Damaster blaptoides* and *Acoptolabrus gehinii* [16,17]. Other predators might feed on snails but there has been little evidence of such [17]. Yet, an earlier study reported drastically different defence strategies [16]: *K. editha* snails hide inside their shell under threat, whereas *K. gainesi* snails swing their shell to prevent the approach of the beetle (figure 1b). Typically, *K. editha* retracted at a constant speed of nearly 1 cm s$^{-1}$, obtained by measuring the length $r$ of exposed snail body at the approach of a beetle. The line on the graph in figure 1b corresponds to a linear regression with a coefficient of correlation $R^2$ of 0.92. On the opposite, *K. gainesi* was found to swing their shells in a periodic manner and at increasing velocity. In figure 1b, we measured the angle $\theta_{swing}$ between the long axis of the shell and the snail body as a function of time, at the approach of the beetle. Similar measurements were conducted for the seven repetitive swings of the shell that the snail had to perform before the beetle moved away (see electronic supplementary material, figure S1). Each swing could be fitted with a cosine function, as illustrated by the black line on the graph in figure 1b, with coefficients of correlation $R^2$ of 0.88–0.99. As the shell plays a major role in those two defence behaviours, it can be expected that their respective shell geometry, mechanics and internal structure differ.

Indeed, alongside this contrasting behavioural response, the two species exhibited different phenotypes, presumably adapted to their defence mechanisms. Overall, *K. editha* presented small shells with an average diameter $d$ and height $l$ of $1.9 \pm 0.06$ cm and $1.6 \pm 0.06$ cm, respectively, when compared with $2.9 \pm 0.12$ cm and $2.7 \pm 0.12$ cm for *K. gainesi* (figure 1c,d). We represented those dimensions using the spire index $s = d/l$ in figure 1e [25]. Both species had low-spire-index shells, meaning s < 1. In addition, the shells from *K. editha* had an overall density of $0.4 \pm 0.02$ g cm$^{-3}$, and the shells from *K. gainesi*, of $0.13 \pm 0.01$ g cm$^{-3}$ (figure 1e). Both the spire index and the density were significantly different between the two species, with *p*-values less than 0.005. From these considerations, the shells from *K. gainesi* seem more favourable for a club-like design where a lightweight material is desired along with a large area. Finally, another phenotypic feature is the difference in colour between the two shells, with *K. gainesi* being brown, and *K. editha*, white.

In addition, the orientation of the columella in the two species also differed (figure 2). The columella is the direction of high mechanical strength in the shell [26]. Hence, the shell will be able to sustain high mechanical loadings when the load is applied along the direction of the columella. The columella can be defined by two angles, the angle at rest, $\theta_{rest}$, and the angle in action, $\theta_{act}$. $\theta_{rest}$ is the angle between the columella and the aperture, whereas $\theta_{act}$ is the angle between the columella and the horizontal plane (figure 2a,b). During growth, the expansion and the coiling of the tube forming the shell from the

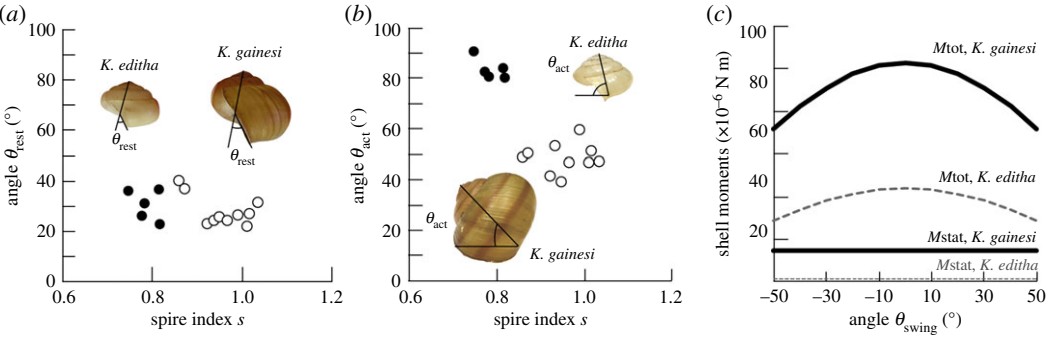

**Figure 2.** Angles at rest $\theta_{rest}$ (a) and in action $\theta_{act}$ (b) as a function of the spire index s for the shells considered in the study (black, K. editha and white, K. gainesi). The pictures illustrate how the angles were measured. (c) Calculated moments of the shells when the snail is on a horizontal surface (Mstat) and if the snail is swinging its shell (Mtot) for K. editha (grey and dashed) and K. gainesi (black and full).

accretion of material at the aperture, happen simultaneously [27]. There is thus an intimate correlation between the growth and the final shape of the shell. In our measurements, we found that the difference in $\theta_{rest}$ was not highly significant, with a p-value of 0.5 only (figure 2a). This could suggest that the shells of the two species followed similar formation path. Studies on live species would be needed to investigate this point further.

However, we found a highly significant difference in the angles $\theta_{act}$ with a p-value less than 0.005. In live specimens, the columella of K. editha was mostly oriented at 90° from the horizontal, whereas K. gainesi had its columella oriented around 50°. In this study, the columella's orientation on live snails could not be measured due to the absence of live specimens. Nonetheless, we noted that positioning the shells in their equilibrium position on a flat surface could approximate the natural orientation of the columella well enough. We, therefore, measured $\theta_{act}$ of $83 \pm 2°$ for K. editha and $48 \pm 3°$ for K. gainesi, which are close to the observations on live snails (figure 2b). The tilt of the columella is known to have adapted for the best stability of the snail during its motion [25]. One of the two species has thus its shell in a lesser stable position.

To determine the stability of the shells on the snails, on a horizontal surface, we calculated the shell moment Mstat using

$$M\text{stat} = \rho V g \frac{l}{4} \cos \theta_{act}, \tag{3.1}$$

where $\rho$ is the density of the shell, V its volume, g the gravitational acceleration constant and l the height (see schematics in electronic supplementary material, figure S2). The shell moment describes the rotation of the shell around the columella, under an applied force: gravity when the snail is at rest, or the muscle force of the snail during swinging. The calculated moments were higher for K. gainesi, suggesting a higher shell stability for K. editha (figure 2c). We also calculated the moments generated when the snail is swinging its shell using

$$M\text{tot} = M\text{stat} + \rho V g l \cos \theta_{swing}, \tag{3.2}$$

where $\theta_{swing}$ is the swinging angle as indicated in figure 1b. We made the same calculation for the two species to observe which one would generate the highest moment. Indeed, in the case of an active defence behaviour, there is a need to achieve large moments to move the shell at a high speed and to hit or distract the predator with a higher force. In this respect, K. gainesi is better equipped than K. editha to generate large moments. Thus, there must be a trade-off between the shell stability at rest and the generated moments in snails exhibiting active defence. One can thus hypothesize that in the course of evolution, K. gainesi did not go for high stability, as most snails, but explored another strategy to make use of this imperfect balance.

It can be concluded from this macroscopic characterization of the shells of the two snail species that different phenotypes correlate well with the different behaviour. Indeed, the vertical orientation of the strongest direction, and the high density and compactness of the shells of K. editha are reminiscent of a protective shield. By contrast, the tilting of the strong direction axis, the lightweightness and large dimensions of the shells of K. gainesi draw similarities with a hitting club. In this case, the large radius of the shell also contributes in creating a lever of length approximately the diameter of the

shell. If one assumes a similar muscle strength in both species, *K. gainesi* snails need to supply a force 1.5 times lower than *K. editha* to produce the same work (work = force × lever), since the shell diameter of *K. gainesi* is 1.5 times higher than that of *K. editha*. It is, therefore, easier for *K. gainesi* to move its shell than for *K. editha*. However, to confirm this hypothesis, further investigation at the muscle level is required.

*Karaftohelix editha* and *K. gainesi* exhibit two drastically different strategies in response to predatory attacks that seem to be enabled through a different macroscopic geometry of their shell. Next, we question how this difference in behaviour is also reflected in the mechanical properties of the shells at the microscopic level.

## 3.2. Contrasting mechanics at the microlevel

The morphological and macroscopic characterizations allow us to hypothesize that *K. editha*'s shells should be hard and stiff to resist indentation and crushing when the snail body is retracted, whereas *K. gainesi*'s shells should be tough and able to resist the dynamic stresses that would occur during swinging. To verify those hypotheses, we first investigated the morphology of cracks and other damage at the surface of the shells. Then, we performed nanoindentation on polished cross-sections of the shells on a part located at *ca* 5 mm from the aperture (figure 3).

All specimens collected were observed under optical microscope to carryout a post-mortem analysis. Even if the origin and date of the damage are unknown, their morphology indicated an overall different material's response between the two species. We selected representative images to illustrate best our observations (figure 3*a–f*). First, all damage appeared white on the shells. This whiteness can be attributed to an increase in roughness, causing scattering of the light, as well as possible strain-whitening effects [28] (figure 3*a–e*). *Karaftohelix editha*'s shells generally displayed damage over the entire shell, with the occurrence of sharp and deep indentation pits (figure 3*b*) as well as wear (figure 3*c*). By contrast, *K. gainesi*'s shells were damaged mostly around the protoconch at the top of the shell (figure 3*d*). No sharp pits were observed on any of the shells, but round defects were generally present (figure 3*e*). These defects were surrounded by similarly round microcracks showing delamination (figure 3*f*). Usually, the presence of cracks indicates a brittle material [29]. However, in the case of *K. gainesi*, those microcracks are tortuous (figure 3*f*, red arrow) and forming closed loops, suggesting crack arrest (figure 3*f*, yellow arrow). In this case, the tortuosity and the unusual circular crack paths strongly support the presence of extrinsic toughening mechanisms that limit the crack propagation and prevent the catastrophic failure of the shell [29]. In the case of *K. editha*, the presence of some sharp pits and wear could be due to a harder composition, with high strength but low wear-resistance.

To complement these macroscopic post-mortem observations, we performed nanoindentation using a diamond Berkovitch tip on polished cross-sections of the shells. Nanoindentation is a method that provides the local Young's modulus of a material, i.e. its stiffness. For both species, the local indentation profiles varied with the position of the indent within the cross-section (figure 3*g,h*). The variation was higher in the case of *K. gainesi*. Indeed, Young's modulus from *K. editha*'s shells ranged from 13 to 15 GPa, whereas *K. gainesi*'s shells had a modulus of 12 GPa on average, with a significant drop to 6 GPa on the outer layer of the shell. These measurements confirm that the shells from *K. editha* are harder and stronger than those of *K. gainesi*, in line with the observation of the sharp surface pits. Furthermore, the presence of a layer with lower modulus on the outer surface of *K. gainesi*'s shells could lead to shock-absorption properties and more toughening.

To further understand this difference in mechanical response, we took a closer look at the microstructure and composition of the shell. Indeed, it is known that natural hard materials are composites with a hierarchical and complex intertwining between mineral and organic components [30]. In the case of snail shells, there are four levels of hierarchy: mineral crystals within an organic matrix (level 1), orientation of groups of crystals in specific patterns (level 2), multiple layers with different orientations (level 3) and all this in a three-dimensional macroscopic shape (level 4).

## 3.3. Hierarchical microstructure of the shells

Natural hard biomaterials are composites made of organic polymers based on proteins and polysaccharides, and inorganic particles. In mechanically resistant biocomposites, those particles are generally anisotropic, in the form of rods or platelets [2,31,32]. During their formation, these anisotropic particles grow with a controlled orientation that influences the mechanical properties of the entire composite [33]. Most common orientation patterns are the brick-and-mortar structure and the columnar phase of calcium carbonate in seashells [31], the plywood arrangements of hydroxyapatite rods in the

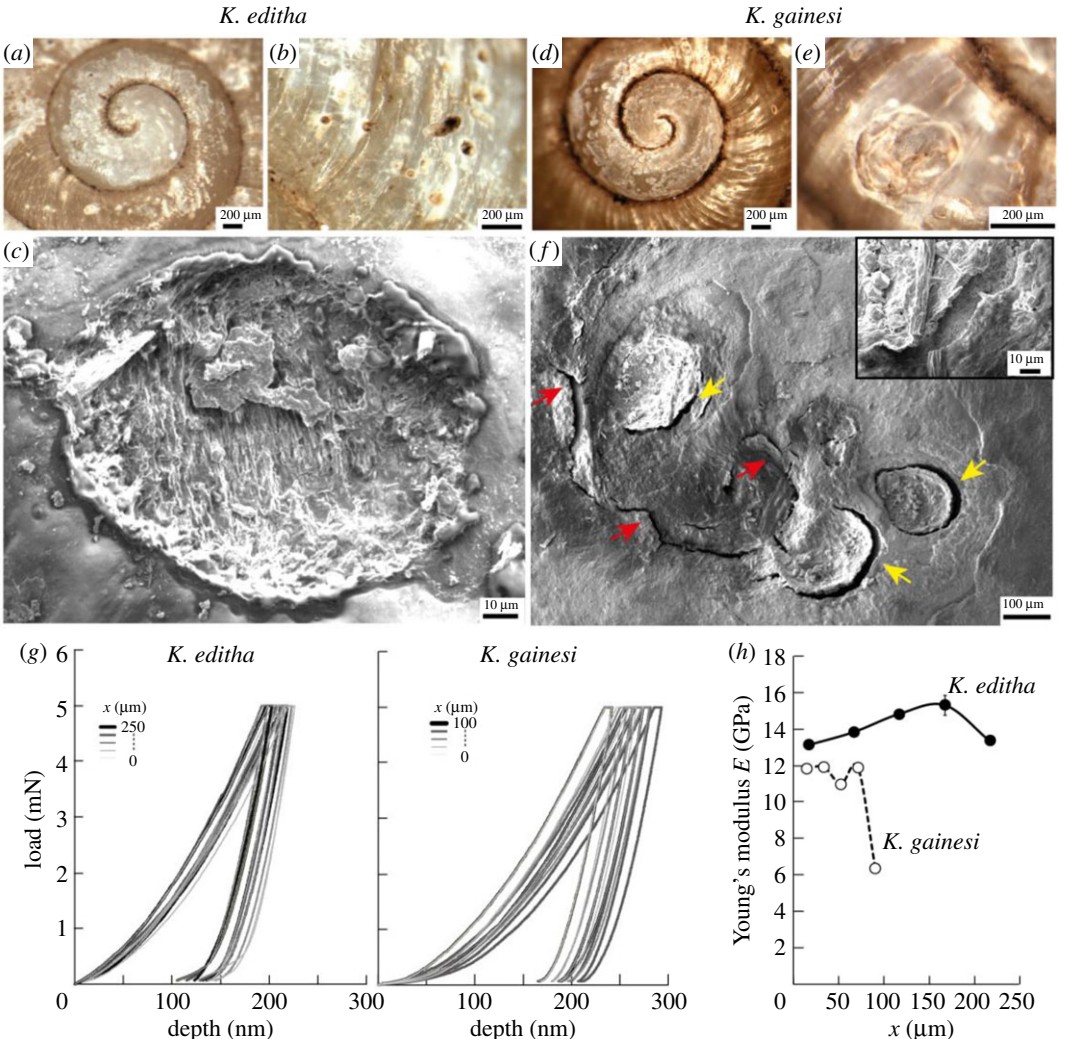

**Figure 3.** Local mechanical response and damage resistance: macroscopic images of damage at the surface of the shells from *K. editha* (*a,b*) and *K. gainesi* (*d,e*) and electron micrographs close-up views (*c*) *K. editha* and (*f*) *K. gainesi*. The insert in (*f*) highlights delamination at the damage site. The red and yellow arrows are pointing at a tortuous cracks and delamination, respectively. (*g*) Load–displacement curves obtained by nanoindentation on polished cross-sections of the shells at different positions *x* along this cross-section (the side in contact with the snail body corresponds to $x = 0$). (*h*) Young's modulus as a function of the position *x* across the thickness of the shell.

mantis shrimp or the cross-lamellar structure in the conch shell [33]. The microstructure in the shells of land snails has been studied extensively [34–37], but the characterization and comparison of those particular two species is still absent from the literature. In the following, we compare and discuss the microstructure and composition of the shells of *K. editha* and *K. gainesi* (figure 4).

To carry out this investigation, we selected two intact specimens from each species and cut stripes close to the aperture. These stripes were then fractured and their cross-section analysed in an electron microscope. First, the thickness of *K. editha*'s shells was higher than that of *K. gainesi*, with values $Xt$ of $227 \pm 5$ µm and $85 \pm 3$ µm, respectively (figure 4*a*). This difference in thickness, in addition to the variation in overall macroscopic dimensions, probably accounts for the lightweightness of *K. gainesi*'s shells over *K. editha*'s. Similar thicknesses were measured during the nanoindentation experiments (figure 3*h*).

Furthermore, *K. gainesi*'s shells presented an organic outer layer, presumably the periostracum, that was not present in *K. editha*'s shells (figure 4*b,c*). This organic layer would explain the drop in Young's modulus in the nanoindentation profile of *K. gainesi* at $x = 100$ µm (figure 3*h*). One of the main functions of such organic layer is in the active repair of the shell structure by promoting biomineralization events to occur [38–41]. For example, Charrier *et al.* recorded the presence of an organic layer in *Notodiscus hookeri*, highly rich in proteins and deprived of chitin, associated with soft mechanical properties [39]. The authors also observed that lesser mineralization in the shells led to thicker organic layers and larger

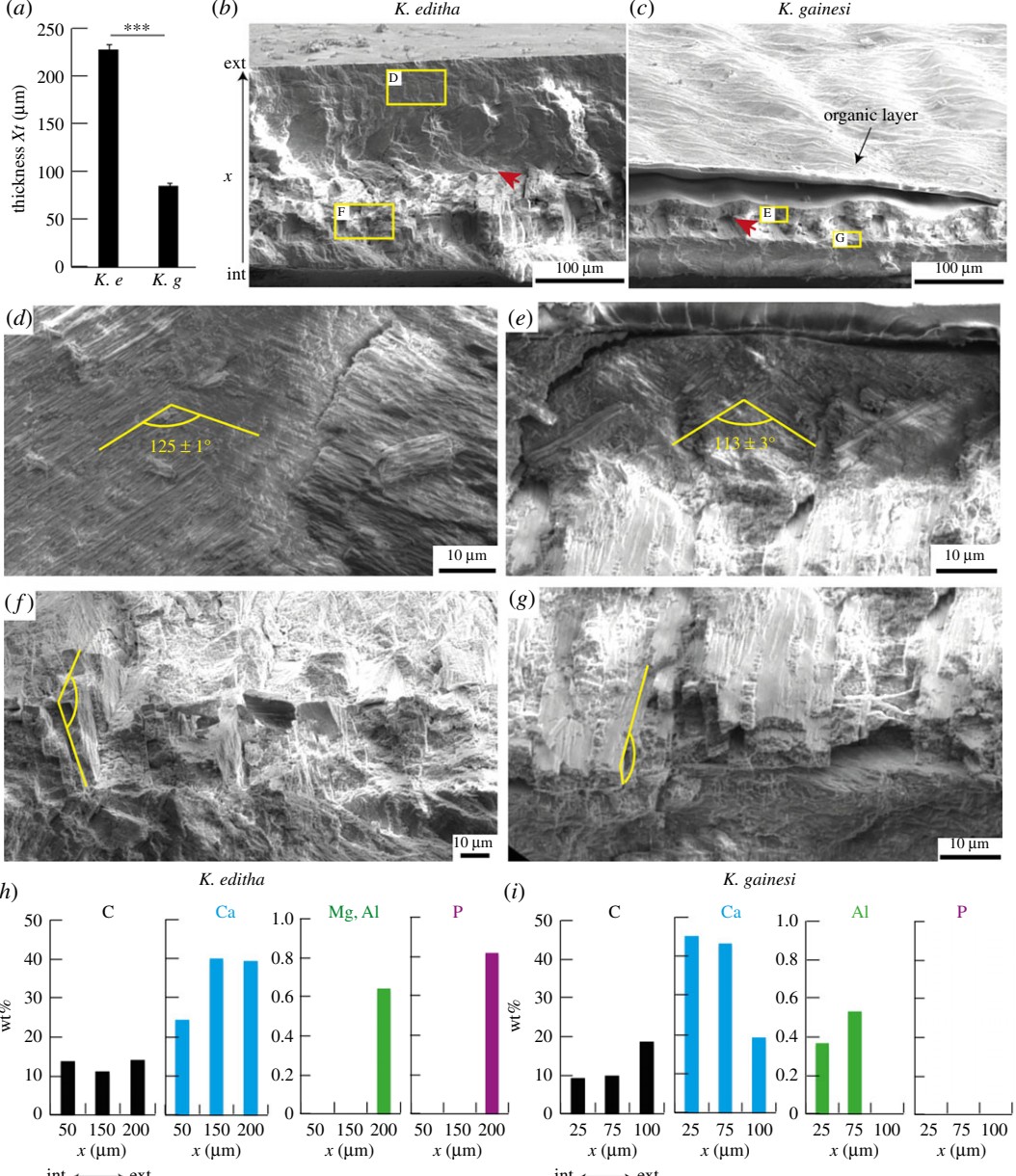

**Figure 4.** (*a*) Thickness of the shell wall of both species, measured at the same distance from the aperture. Electron micrographs of fractured cross-sections of the shells of *K. editha* (*b,d,f*) and *K. gainesi* (*c,e,g*). (*b,c*) The entire cross-section, (*d,e*) the cross-lamellar arrangement close to the exterior side and (*f,g*) the cross-lamellar arrangement on the interior side. The yellow lines indicate the lamellae criss-crossing at approximately 125° and approximately 113°. The red arrows indicate crack deflection as the lamellae orientation changes at the interface between layers. (*h,i*) Elemental composition in wt% throughout the thickness $x$ of the shells, for *K. editha* (*h*) and *K. gainesi* (*i*). The composition was obtained using energy-dispersive X-ray analysis (EDX) (full spectra in electronic supplementary material, figure S3).

shell dimensions. They hypothesized that the decreased energy required for the biomineralization could be used instead in those shells to increase the overall dimensions. Based on those observations, we suggest that in the case of *K. gainesi*, the energy that was not used to build a thick and highly mineralized shell was input into the synthesis of a larger shell, a larger aperture and in the snail's columellar muscle. However, other studies on the contrary argue that a shell containing a high content in protein is more demanding in terms of energy [42]. In the absence of evidence that mineralization costs more energy than the proteinic coating, one can nevertheless conclude that there is a trade-off between growing large shells or highly mineralized ones. Finally, having an organic external layer could participate in the self-healing of the shell as well as in the dampening of stresses occurring after repetitive shocks. Those two mechanisms would benefit greatly *K. gainesi*.

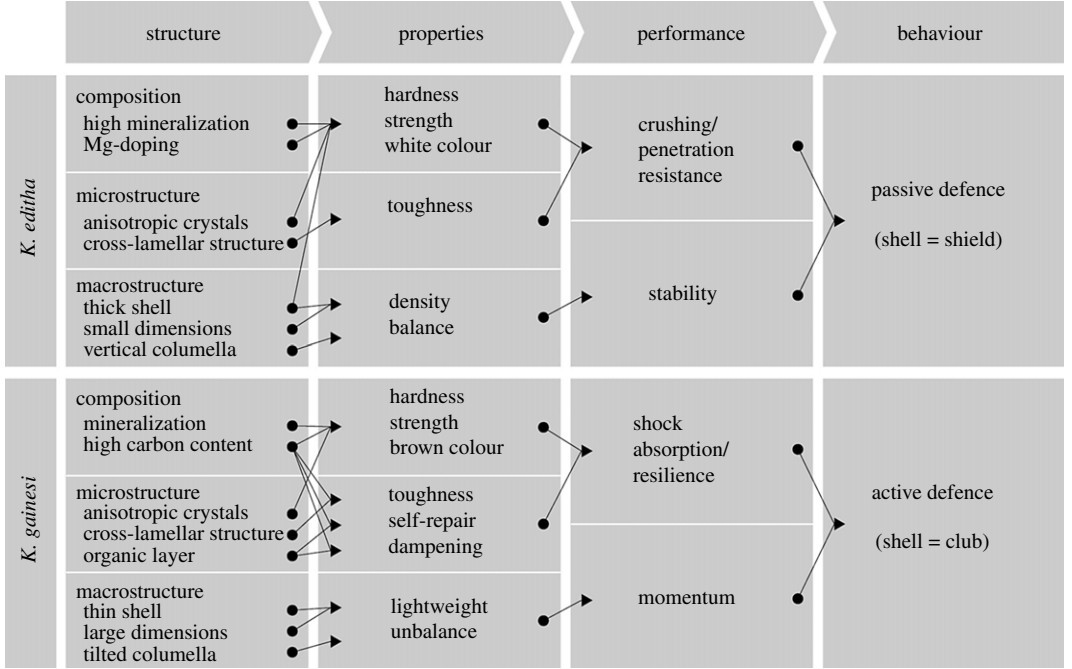

**Figure 5.** Correlations between the shell structure, mechanical properties, functions and the snail defence behaviour for *K. editha* and *K. gainesi*.

Below the periostracum, if any, the internal microstructure of the both shells displayed anisotropic crystals of calcium carbonate that are oriented along various patterns. Overall, both species exhibited the same microscopic organization in three layers with cross-lamellar arrangements [39]. The first external mineralized layer presented elongated mineral crystals criss-crossing at an angle of 113°–125° (figure 4*d*,*e*). The middle layer presented larger mineral crystals stacked in prisms in similar cross-lamellar microstructure, but with a change in orientation of the lamellae (figure 4*f*,*g*, yellow lines) [43]. The change of lamellae orientation with respect to the top layer deflected the fracture induced during the sample preparation, rending this middle layer more visible (see red arrows on figure 4*b*,*c*). Finally, the third layer, at the interior of the shell, appeared without apparent microstructure suggesting an amorphous phase or crystallites in random orientations.

Finally, we characterized the chemical composition of the shells by energy-dispersive X-rays in the three layers in thickness of the shells (see electronic supplementary material, figure S3 for the entire spectra) (figure 4*h*,*i*). Figure 4 shows representative results. Globally, the shells of *K. editha* and *K. gainesi* were composed of the same elements, namely C, O and Ca as major constituents and Mg, P and Al in trace amounts. P was present in both species but in lower concentrations in the case of *K. gainesi*. Furthermore, the two species had distinct relative concentrations in Mg and Al. Indeed, Mg was only found within *K. editha*'s shells, whereas *K. gainesi* had a larger amount of Al. In particular, Mg ions in trace amounts are usually responsible for increased Young's modulus [44,45]. As expected from the microscopic observations, the outer layer of *K. gainesi*'s shells presented a high concentration of carbon with low mineral content, confirming the presence of a predominantly organic layer on the outer surface. By contrast, *K. editha*'s shells had a high concentration of minerals, in particular, Ca, Mg and P, in the outer layer. This high concentration probably increased the hardness and Young's modulus of the shell and provided the white colour of the shell. The organic layer in *K. gainesi*, rich in carbon, leads to a darker colour that could hypothetically increase the camouflage of the snail on the ground.

## 4. Conclusion

The two genetically related species *K. editha* and *K. gainesi* have distinct response to predatory attacks despite sharing the same habitat and similar genes. These distinct behaviours were correlated with contrasting structures of their shell, from the macroscopic morphology down to the microstructure and chemical composition. The structures were associated with the mechanical responses at the microscopic level, further impacting the mechanics and performance at the macroscopic level (figure 5). We observed that

*K. editha* had white shells presenting high hardness, strength and toughness. These properties should provide for crushing and penetration resistance. Also, the relatively high density combined with the vertical orientation of the columella, could enable the shells to be well balanced, providing stability to the snail. These characters make the shells well suited for the passive defence mechanism where the shell performs as a shield against predation. The shells of *K. gainesi* also presented hardness and strength but at a lower degree than those of *K. editha*, whereas their toughness could be enhanced by the organic periostracum. This proteinic layer could also be expected to contribute to self-healing, shock absorption and dampening properties, as well as a darker colour to use as camouflage. Contrary to *K. editha*, the tilting of the angle of the columella and the lower density of *K. gainesi*'s shells make them more lightweight and unbalanced, in turn providing momentum. This momentum presumably allows the snail to move its shell with a lower muscle force as compared to the case of a smaller, heavier and well-balanced shell. With these considerations, the shell of *K. gainesi* seems to be better suited for the active defence mechanism where it is used as a club. One can thus postulate that the divergence in defence behaviour results from a trade-off between mechanical properties and the energy required for swinging. Although we listed correlations between the structure of the shells and the behaviour of snail, we cannot predict from our analysis if the shell evolved first, then the behaviour, or the opposite, or if these changes occurred concomitantly. It is, however, legitimate to assume that each structure has evolved to be optimized for each defence scenario.

Considering the large diversity of land snail species across the globe, it remains puzzling that only a few were found to exhibit an active defence behaviour. To the best of our knowledge, only two *Karaftohelix* species have been reported to have such active defence [16]. *A priori*, the coexistence of species with passive and active response should indicate a similar efficiency of the strategy for survival in their habitat. One reason for this special case in Hokkaido Island might be the large prevalence of beetle-type predators, whereas birds and mammals also feed on snails in other parts of the world [17]. Carabid beetles have two ways of preying on snails, by crushing their shells or by entering the shell through the aperture [46]. Further information on the beetle's strategy in the presence of these snails would be needed to complement our understanding of predator–prey interactions. Although the principle of adding a soft layer onto a hard and more brittle composite is not new in engineering, the present study illustrates well how multiscale structures are critical to the functions and performance of macroscopic objects. It is anticipated that further understanding of the structure-properties–performance relationships of biomaterials would contribute to evolutionary ecology as well as to biomimetic engineering. Although it is challenging to draw general conclusions when comparing only two species, this study illustrates how two structures with similar ultimate goal can be designed in different ways to realize different specific functions. In particular, how a structure can integrate the environmental constraints to achieve the best strategy is interesting in view of minimizing the cost and complexity in synthetic materials parts.

Data accessibility. The data that support the findings of this study are presented in the article. Additional data are available in the electronic supplementary material.

Authors' contributions. H.L.F and Y.M. designed the study, Y.M. collected the specimens and studied their behaviour, H.L.F. conducted the materials characterization and wrote the original draft of the paper. H.L.F. and Y.M. discussed the results and their implications at all stages and revised the manuscript.

Competing interests. We declare we have no competing interests.

Funding. Funding from the Swiss National Science Foundation (grant no. P2EZP2_172169) and start-up grant from Nanyang Technological University (M4082382.050) are gratefully acknowledged.

Acknowledgements. The authors acknowledge the Facilities for Analysis, Characterisation, Testing and Simulations (FACTS) at NTU for access to the electron microscopy facilities, Ali Miserez for allowing us to use the nanoindenter and Wing Chung Liu for proof-reading.

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
