## [Reviewer comments · Royal Society Open Science]

Review History

RSOS-191471.R0 (Original submission)

Review form: Reviewer 1

Is the manuscript scientifically sound in its present form?

No

Are the interpretations and conclusions justified by the results?

No

Is the language acceptable?

Yes

Do you have any ethical concerns with this paper?

No

Have you any concerns about statistical analyses in this paper?

Yes

Recommendation?

Reject

Comments to the Author(s)

Despite some improvements of the revised manuscript entitled "Structure-behaviour relationships of the shells from two genetically closely related snail species", the manuscript suffers from missing or incorrect assumptions.

Materials and methods

Are the shells collected empty, with dead animals???? How was the soft tissues removed from the shells? What about the taxonomy of the animals? Illustrations showing the different faces (apex, aperture...) of the shells are necessary for people not familiar with gastropods.

Thanks for this question, we have added the details about the preparation of the shells in the manuscript, along with an additional reference. We think, however, that the manuscript is already lengthy enough and that the reader can look at our references should he/she need more general information on gastropods.

Such figures and other details can be added in the supplementary data.

I am not sure that non-familiar readers have time to look for references to understand the paper.

"Polishing, nevertheless, was used for the nanoindentation testing. We added a few details as well:

"Nanoindentation was carried out on polished cross-sections of the shell. In summary, pieces of shells that were cut at similar locations as observed in SEM were embedded in epoxy (Specifix, Struers, Germany), and mirror-polished using SiC papers of decreasing roughness and finally colloidal diamond suspensions. Polishing debris were removed using an ultrasonication bath."

Was the epoxy inclusion made at room temperature? At atmospheric atmosphere or using autoclave?

Using water for ultrasonic bath? How long?

Some colloidal polishing products are in water, some are in oil. So my request.

"We acknowledge that Pt sputtering might not be the best choice to observe light elements, however, this is the case for thin films or under very low voltages. In our case, the Pt coating consisted only of 5 to 10 nm at most, whereas the electron beam had 5 keV. The signal from the Pt coating therefore was too little and could not even be recorded by EDX, as shown on the spectra in the Supplementary Information. One could bring the additional argument that coating with C will also prevent us to analyze the location and presence of carbon in the shell. The protocol we use is the standard protocol for element characterization. Among the plethora of publications that use this protocol, we invite the reviewer to look at that one, for example: Multi-scale structural design and biomechanics of the pistol shrimp snapper claw, Amini S, Tadayon M, Chua I, Miserez, A, *Acta Biomaterialia* 73 (2018) 449- 457.

Despite the claim of the authors, Pt coating is not the best. The normal set up conditions for chemical EDS analyses are a C coating, at 15 keV. C coating does not prevent to see C in the samples, because the C coating layer is uniform, and C in the shell is not hidden. Moreover, there is not data about the spot size, the dwell time.....

"Periostraci in gastropods' shells have been observed in many other species and have been shown to play an active part in the growth and in the repair of the shell, playing a similar role than the eggshell in avian eggs"

I do not yet understand this sentence; periostracum is an essential piece of the normal biomineralization process, and is not used only for repair. As for the comparison with the eggshell, it is very inappropriate because it is an experiment the authors of which are aware of the

difference between molluscs and aves, as shown by the title of the paper: "...as a template of foreign scaffold..."

The chemical analyses are very poorly done.

"We followed a standard method that was well-suited to our samples."

A standard method is used for all samples!

Despite the potential interest of the manuscript, the analyses are not well done, so that it is difficult to be confident about the results. To show that the structure is hierarchical is verbose, because all organisms show a hierarchical organization: cells contains organs and nucleus, organs are made of cells, body is made of organs... Every mineralized tissue has a hierarchical arrangement: nanorods in enamel prisms, prisms in layers....

Review form: Reviewer 2

Is the manuscript scientifically sound in its present form?

Yes

Are the interpretations and conclusions justified by the results?

Yes

Is the language acceptable?

Yes

Do you have any ethical concerns with this paper?

No

Have you any concerns about statistical analyses in this paper?

No

Recommendation?

Accept with minor revision (please list in comments)

Comments to the Author(s)

This new version has been improved regard to the comments of the reviewers.

It remains some informations that do not appear in this new text. Actually a paper must be understandable in its own form and not besides a previous paper : it does not take several lines to add some informations that strengthen the results. It is very important to understand that the experiment correspond to a "real" prey-predator situation to avoid bias. Especially, some informations that can be found in the ref 16 will gain to figure in this text.

- Origin of the shell: You answer to my question about the localities that all the locations have been indicated in ref 16 (Morii et al., 2016). But in the ref 16 there were more than 30 shells listed.

And you have measured 6 shells of *K. editha*. Please indicate from where they are coming from.

- The fact the both predators come from the same locality and the both prey species come from the same locality also must be indicated.

- It lacks information about the videos/movies analysed, please indicate the number of the movies found in ref 16 (Morii et al., 2016) or better the link.

I agree with the referee 2 concerning the size of the figure 1 but for the right part B (very-too small). One observation concerning images of *K. gainesi*: the predator does not change its position during the rotation of the shell whereas in movie of the ref 16, the predator moves. Is

there an adaption of the shell position regard to the position of the predator (not seen by the prey because the eyes are located at the opposite side)

In the conclusion, the Figure 5 has not been more explained or justified by other data, actually the data inside are partial regard to the subject and do not allow the links (correlations?) shown (the mechanical strength, the insertion of muscles and many other things have to be studied). I suggest again to remove it or at least to remove the arrows. In a "table" format with simple columns, the different characteristics appear as a summary of the results and not as an attempt to establish relationships (the legend has to be changed).

Minor comments

There is no more cone illustration in figure 1C. Change the text line 52 p6 (and line 56- p.8?)

P6 line 20-21: precise (if correct) that apparently (?) there is no preference or specificity between the two prey-predators.

P7 line 57: "similar shell development and formation": at which level? What is the difference between development and formation?

Line 11 p2 & Line 17 p 13 Minus character for mantis shrimp.

Line 11 p13 these and not theses

Line 24 p13 specimens and not specimen

Review form: Reviewer 3

Is the manuscript scientifically sound in its present form?

Yes

Are the interpretations and conclusions justified by the results?

Yes

Is the language acceptable?

Yes

Do you have any ethical concerns with this paper?

No

Have you any concerns about statistical analyses in this paper?

No

Recommendation?

Accept with minor revision (please list in comments)

Comments to the Author(s)

The authors present interesting results from a comparison of two closely-related terrestrial gastropod species that exhibit different predator avoidance strategies. They found that *K. editha* that retreats into its shell when confronted with beetle predators has a harder more stable shell and that *K. gainensi* that swings its shell to club the predator has a more lightweight shell that allows for greater movement. These shell characteristics would seem to make sense with regard to their predator avoidance strategies. I do have a few suggestions, comments, and questions.

1. Page 2, line 18. The word "and" should be inserted between "patterns" and "chirality."
2. Page 2, line 22. "divergence" should be "divergences."

3. Page 2, line 59. Are the two species that show the behavioral response sister species or did this trait evolve more than once? Has a phylogeny of this genus been done? This would be interesting to know.

4. Page 4, line 9. I think a better wording would be "In total, 6 shells from living adult *Karafkaohelix editha* and 11 from living adult . . ." Then remove the words "alive" from the next sentence.

Page 5, Figure 1D. It is not clear in this figure which set of points is which species. This should be defined in the figure heading.

Page 7. Do the angles described in figure 2 change during development? I wonder if the differences in strategy may in part be explained by the differences in adult size.

Page 8, line 23. I think an explanation of "shell moment" would be beneficial to many readers.

Page 8, line 49. The authors suggest that the snails evolved their shell morphology first and their defense strategy second. Could the reverse be true. Could the changes in shell be selected for due to difference in defense strategy?

Page 11, line 13. I think a definition and explanation of the importance of Young's modulus is needed here.

Page 13, line 28. The authors use the term "significantly" to describe difference in shell thickness. The term significant in science is very much tied to statistics. No statistics were done to compare the thicknesses that I can see, so I suggest using a different word such as "much" or "substantially."

Page 14, lines 12-31. The authors discuss the differences in the organization of the shell between the species. Two snails were chosen for analysis for each species. Was any variation found within species?

Page 15, Conclusion. My primary concern with this paper is that the authors show a correlation between predator avoidance strategy and morphology based on only two species. One could make a much stronger case if multiple species were examined representing each strategy. The authors state that there are two species that exhibit the behavioral response and multiple other species that retract into their shells when confronted with the predator. It is dangerous to make conclusions based on a sample size of 2. I realize that this paper represents a substantial amount of work. I suggest, at the very least, that the authors acknowledge the limitation and danger of basing conclusions on just two species.

Decision letter (RSOS-191471.R0)

29-Nov-2019

Dear Dr LE FERRAND,

The editors assigned to your paper ("Structure-behaviour relationships of the shells from two genetically closely related snail species") have now received comments from reviewers. We would like you to revise your paper in accordance with the referee and Associate Editor suggestions which can be found below (not including confidential reports to the Editor). Please note this decision does not guarantee eventual acceptance.

Please submit a copy of your revised paper before 22-Dec-2019. Please note that the revision deadline will expire at 00.00am on this date. If we do not hear from you within this time then it will be assumed that the paper has been withdrawn. In exceptional circumstances, extensions may be possible if agreed with the Editorial Office in advance. We do not allow multiple rounds of revision so we urge you to make every effort to fully address all of the comments at this stage. If deemed necessary by the Editors, your manuscript will be sent back to one or more of the original reviewers for assessment. If the original reviewers are not available, we may invite new reviewers.

- Data accessibility

If you wish to submit your supporting data or code to Dryad (<http://datadryad.org/>), or modify your current submission to dryad, please use the following link:
<http://datadryad.org/submit?journalID=RSOS&manu=RSOS-191471>

- Competing interests

- Authors' contributions

AB carried out the molecular lab work, participated in data analysis, carried out sequence alignments, participated in the design of the study and drafted the manuscript; CD carried out the statistical analyses; EF collected field data; GH conceived of the study, designed the study,

coordinated the study and helped draft the manuscript. All authors gave final approval for publication.

- Acknowledgements

- Funding statement

Kind regards,

Lianne Parkhouse

Editorial Coordinator

on behalf of the Associate Editor, and Professor Kevin Padian (Subject Editor)

Associate Editor's comments:

The reviewers of this iteration of the paper indicate you've made good faith efforts to improve your manuscript; however, a number of concerns remain, and we invite you to address these thoroughly. A number of key concerns involve the degree to which your conclusions can be extrapolated to other scenarios - given the sample size limitations, we would like you to more fully address this concern - you should, as one of the reviewers observes, make greater efforts to address the limitations of the study. Additionally, some comments regarding the written English have been made. You should seek the advice of a native speaker of English or use a service such as those at <https://royalsociety.org/journals/authors/language-polishing/> before resubmitting.

Reviewers' Comments to Author:

Reviewer: 1

Comments to the Author(s)

Despite some improvements of the revised manuscript entitled "Structure-behaviour relationships of the shells from two genetically closely related snail species", the manuscript suffers from missing or incorrect assumptions.

Materials and methods

Are the shells collected empty, with dead animals???? How was the soft tissues removed from the shells? What about the taxonomy of the animals? Illustrations showing the different faces (apex, aperture...) of the shells are necessary for people not familiar with gastropods.

Thanks for this question, we have added the details about the preparation of the shells in the manuscript, along with an additional reference. We think, however, that the manuscript is already lengthy enough and that the reader can look at our references should he/she need more general information on gastropods.

Such figures and other details can be added in the supplementary data.

I am not sure that non-familiar readers have time to look for references to understand the paper.

“Polishing, nevertheless, was used for the nanoindentation testing. We added a few details as well:

“Nanoindentation was carried out on polished cross-sections of the shell. In summary, pieces of shells that were cut at similar locations as observed in SEM were embedded in epoxy (Specifix, Struers, Germany), and mirror-polished using SiC papers of decreasing roughness and finally colloidal diamond suspensions. Polishing debris were removed using an ultrasonication bath.”

Was the epoxy inclusion made at room temperature? At atmospheric atmosphere or using autoclave?

Using water for ultrasonic bath? How long?

Some colloidal polishing products are in water, some are in oil. So my request.

“We acknowledge that Pt sputtering might not be the best choice to observe light elements, however, this is the case for thin films or under very low voltages. In our case, the Pt coating consisted only of 5 to 10 nm at most, whereas the electron beam had 5 keV. The signal from the Pt coating therefore was too little and could not even be recorded by EDX, as shown on the spectra in the Supplementary Information. One could bring the additional argument that coating with C will also prevent us to analyze the location and presence of carbon in the shell. The protocol we use is the standard protocol for element characterization. Among the plethora of publications that use this protocol, we invite the reviewer to look at that one, for example: Multi-scale structural design and biomechanics of the pistol shrimp snapper claw, Amini S, Tadayon M, Chua I, Miserez, A, *Acta Biomaterialia* 73 (2018) 449- 457.

Despite the claim of the authors, Pt coating is not the best. The normal set up conditions for chemical EDS analyses are a C coating, at 15 keV. C coating does not prevent to see C in the samples, because the C coating layer is uniform, and C in the shell is not hidden. Moreover, there is not data about the spot size, the dwell time.....

“Periostraci in gastropods' shells have been observed in many other species and have been shown to play an active part in the growth and in the repair of the shell, playing a similar role than the eggshell in avian eggs”

I do not yet understand this sentence; periostracum is an essential piece of the normal biomineralization process, and is not used only for repair. As for the comparison with the eggshell, it is very inappropriate because it is an experiment the authors of which are aware of the difference between molluscs and aves, as shown by the title of the paper: “....as a template of foreign scaffold...”

The chemical analyses are very poorly done.

“We followed a standard method that was well-suited to our samples.”

A standard method is used for all samples!

Despite the potential interest of the manuscript, the analyses are not well done, so that it is difficult to be confident about the results. To show that the structure is hierarchical is verbose, because all organisms show a hierarchical organization: cells contains organs and nucleus, organs are made of cells, body is made of organs... Every mineralized tissue has a hierarchical arrangement: nanorods in enamel prisms, prisms in layers....

Reviewer: 2

Comments to the Author(s)

This new version has been improved regard to the comments of the reviewers.

It remains some informations that do not appear in this new text. Actually a paper must be understandable in its own form and not besides a previous paper : it does not take several lines to add some informations that strengthen the results. It is very important to understand that the

experiment correspond to a “real” prey-predator situation to avoid bias. Especially, some informations that can be found in the ref 16 will gain to figure in this text.

- Origin of the shell: You answer to my question about the localities that all the locations have been indicated in ref 16 (Morii et al., 2016). But in the ref 16 there were more than 30 shells listed. And you have measured 6 shells of *K. editha*. Please indicate from where they are coming from.
- The fact the both predators come from the same locality and the both prey species come from the same locality also must be indicated.
- It lacks information about the videos/movies analysed, please indicate the number of the movies found in ref 16 (Morii et al., 2016) or better the link.

I agree with the referee 2 concerning the size of the figure 1 but for the right part B (very-too small). One observation concerning images of *K. gainesi*: the predator does not change its position during the rotation of the shell whereas in movie of the ref 16, the predator moves. Is there an adaption of the shell position regard to the position of the predator (not seen by the prey because the eyes are located at the opposite side)

In the conclusion, the Figure 5 has not been more explained or justified by other data, actually the data inside are partial regard to the subject and do not allow the links (correlations?) shown (the mechanical strength, the insertion of muscles and many other things have to be studied). I suggest again to remove it or at least to remove the arrows. In a “table” format with simple columns, the different characteristics appear as a summary of the results and not as an attempt to establish relationships (the legend has to be changed).

Minor comments

There is no more cone illustration in figure 1C. Change the text line 52 p6 (and line 56- p.8?)

P6 line 20-21: precise (if correct) that apparently (?) there is no preference or specificity between the two prey-predators.

P7 line 57: “similar shell development and formation”: at which level? What is the difference between development and formation?

Line 11 p2 & Line 17 p 13 Minus character for mantis shrimp.

Line 11 p13 these and not theses

Line 24 p13 specimens and not specimen

Reviewer: 3

Comments to the Author(s)

The authors present interesting results from a comparison of two closely-related terrestrial gastropod species that exhibit different predator avoidance strategies. They found that *K. editha* that retreats into its shell when confronted with beetle predators has a harder more stable shell and that *K. gainesi* that swings its shell to club the predator has a more lightweight shell that allows for greater movement. These shell characteristics would seem to make sense with regard to their predator avoidance strategies. I do have a few suggestions, comments, and questions.

1. Page 2, line 18. The word "and" should be inserted between "patterns" and "chirality."

2. Page 2, line 22. "divergence" should be "divergences."

3. Page 2, line 59. Are the two species that show the behavioral response sister species or did this trait evolve more than once? Has a phylogeny of this genus been done? This would be interesting to know.

4. Page 4, line 9. I think a better wording would be "In total, 6 shells from living adult *Karftohelix editha* and 11 from living adult . . ." Then remove the words "alive" from the next sentence.

Page 5, Figure 1D. It is not clear in this figure which set of points is which species. This should be defined in the figure heading.

Page 7. Do the angles described in figure 2 change during development? I wonder if the differences in strategy may in part be explained by the differences in adult size.

Page 8, line 23. I think an explanation of "shell moment" would be beneficial to many readers.

Page 8, line 49. The authors suggest that the snails evolved their shell morphology first and their defense strategy second. Could the reverse be true. Could the changes in shell be selected for due to difference in defense strategy?

Page 11, line 13. I think a definition and explanation of the importance of Young's modulus is needed here.

Page 13, line 28. The authors use the term "significantly" to describe difference in shell thickness. The term significant in science is very much tied to statistics. No statistics were done to compare the thicknesses that I can see, so I suggest using a different word such as "much" or "substantially."

Page 14, lines 12-31. The authors discuss the differences in the organization of the shell between the species. Two snails were chosen for analysis for each species. Was any variation found within species?

Page 15, Conclusion. My primary concern with this paper is that the authors show a correlation between predator avoidance strategy and morphology based on only two species. One could make a much stronger case if multiple species were examined representing each strategy. The authors state that there are two species that exhibit the behavioral response and multiple other species that retract into their shells when confronted with the predator. It is dangerous to make conclusions based on a sample size of 2. I realize that this paper represents a substantial amount of work. I suggest, at the very least, that the authors acknowledge the limitation and danger of basing conclusions on just two species.

Author's Response to Decision Letter for (RSOS-191471.R0)

See Appendix A.

Decision letter (RSOS-191471.R1)

02-Jan-2020

Dear Dr LE FERRAND,

It is a pleasure to accept your manuscript entitled "Structure-behaviour correlations between two genetically closely related snail species" in its current form for publication in Royal Society Open Science. The comments of the reviewer(s) who reviewed your manuscript are included at the foot of this letter.

on behalf of Prof Kevin Padian (Subject Editor)
openscience@royalsociety.org

Associate Editor Comments to Author:
The authors have addressed the issues raised by the referees on sample-size limitations.

Appendix A

Dear editor,

Thank you for this opportunity to revise our manuscript.

Our response to the reviewers can be found below in blue. All additions and changes in the revised manuscript and supplementary information are highlighted by blue underlined text.

We emphasize that the purpose of the study is not solely related to evolutionary studies but has also a significant engineering interest: how the microstructure and composition can impact the final use and performance of a macroscopic part.

We also had the manuscript proof-read to correct for grammatical errors and typos.

We look forward to hearing your decision.

Sincerely,

Hortense Le Ferrand and Yuta Morii

Referee: 1 (previous referee 2)

Comments to the Author(s)

Despite some improvements of the revised manuscript entitled “Structure-behaviour relationships of the shells from two genetically closely related snail species”, the manuscript suffers from missing or incorrect assumptions.

Materials and methods

Are the shells collected empty, with dead animals???? How was the soft tissues removed from the shells? What about the taxonomy of the animals? Illustrations showing the different faces (apex, aperture...) of the shells are necessary for people not familiar with gastropods.

Thanks for this question, we have added the details about the preparation of the shells in the manuscript, along with an additional reference. We think, however, that the manuscript is already lengthy enough and that the reader can look at our references should he/she need more general information on gastropods.

Such figures and other details can be added in the supplementary data.

I am not sure that non-familiar readers have time to look for references to understand the paper.

Our manuscript indeed aims at a broad readership. We have thus put efforts in keeping the number of specialized terms to its minimum. With respect to the taxonomy of the snails, only columella and apertures are discussed in the manuscript. These two terms are now illustrated in the cartoon in Figure 1C.

“Polishing, nevertheless, was used for the nanoindentation testing. We added a few details as well:

“Nanoindentation was carried out on polished cross-sections of the shell. In summary, pieces of shells that were cut at similar locations as observed in SEM were embedded in epoxy (Specifix, Struers, Germany), and mirror-polished using SiC papers of decreasing roughness and finally colloidal diamond suspensions. Polishing debris were removed using an ultrasonication bath.”

Was the epoxy inclusion made at room temperature? At atmospheric atmosphere or using autoclave? Using water for ultrasonic bath? How long? Some colloidal polishing products are in water, some are in oil. So my request.

The epoxy mounting was done in air at room temperature, we specify this in the manuscript in the materials and methods section: “were cold-mounted in epoxy”. We used water in the ultrasonication bath and cleaned the specimen until there was no more debris to be removed. We added this to the paper: “Polishing debris were washed off using water in a ultrasonication bath until all debris were removed.”

“We acknowledge that Pt sputtering might not be the best choice to observe light elements, however, this is the case for thin films or under very low voltages. In our case, the Pt coating consisted only of 5 to 10 nm at most, whereas the electron beam had 5 keV. The signal from the Pt coating therefore was too little and could not even be recorded by EDX, as shown on the spectra in the Supplementary Information. One could bring the additional argument that coating with C will also prevent us to analyze the location and presence of carbon in the shell. The protocol we use is the standard protocol for element characterization. Among the plethora of publications that use this protocol, we invite the reviewer to look at that one, for example: Multi-scale structural design and biomechanics of the pistol shrimp snapper claw, Amini S, Tadayon M, Chua I, Miserez, A, *Acta Biomaterialia* 73 (2018) 449- 457.

Despite the claim of the authors, Pt coating is not the best. The normal set up conditions for chemical EDS analyses are a C coating, at 15 keV. C coating does not prevent to see C in the samples, because the C coating layer is uniform, and C in the shell is not hidden. Moreover, there is not data about the spot size, the dwell time.....

The set-up conditions for chemical EDS analyses have to be selected depending on the sample and the elements to be looked at.

In the literature, various conditions of coatings and voltages have been used to conduct EDS of mollusc shells. For example: Paula et al. used carbon coating at 25 keV [1], Margariti et al. did not used any coatings and analysed their specimen at 1 keV [2], Rasti et al used gold coating and 20 keV [3], Gallagher et al. used gold-palladium coating at 15 keV [4]. We recorded the data for 60 seconds on 3 areas of 100 μm *100 μm in the same regions and averaged the values on those 3 areas. We added these details in the Materials and Methods section: “EDX spectra were recorded for 60 seconds and the data obtained averaged over 3 neighbouring areas of 100 μm *100 μm ”.

- [1] S. M. De Paula and M. Silveira, “Microstructural characterization of shell components in the mollusc *Physa* sp.,” *Scanning*, vol. 27, no. 3, pp. 120–125, 2005.
- [2] C. Margariti, S. Protopapas, N. Allen, and V. Vishnyakov, “Identification of purple dye from molluscs on an excavated textile by non-destructive analytical techniques,” *Dye. Pigment.*, vol. 96, no. 3, pp. 774–780, 2013.
- [3] H. Rasti, K. Parivar, J. Baharara, M. Iranshahi, and F. Namvar, “Chitin from the mollusc chiton: Extraction, characterization and chitosan preparation,” *Iran. J. Pharm. Res.*, vol. 16, no. 1, pp. 366–379, 2017.
- [4] S. M. Gallagher, J. P. Bidwell, and A. M. Kuzirian, “Strontium is Required in Artificial Seawater for Embryonic Shell Formation in Two Species of Bivalve

Molluscs,” in *Origin, Evolution, and Modern Aspects of Biomineralization in Plants and Animals*, R. E. Crick, Ed. Boston, MA: Springer US, 1989, pp. 349–366.

“Periostraci in gastropods' shells have been observed in many other species and have been shown to play an active part in the growth and in the repair of the shell, playing a similar role than the eggshell in avian eggs”

I do not yet understand this sentence; periostracum is an essential piece of the normal biomineralization process, and is not used only for repair. As for the comparison with the eggshell, it is very inappropriate because it is an experiment the authors of which are aware of the difference between molluscs and aves, as shown by the title of the paper: “...as a template of foreign scaffold...”

This part was corrected in the last version of the manuscript and the comparison to the avian eggs removed.

The chemical analyses are very poorly done.

“We followed a standard method that was well-suited to our samples.”

A standard method is used for all samples!

We responded to this above.

Despite the potential interest of the manuscript, the analyses are not well done, so that it is difficult to be confident about the results. To show that the structure is hierarchical is verbose, because all organisms show a hierarchical organization: cells contains organs and nucleus, organs are made of cells, body is made of organs... Every mineralized tissue has a hierarchical arrangement: nanorods in enamel prisms, prisms in layers....

Indeed, most mineralized tissues are hierarchical, the main point of our paper is to study the correlations between the (hierarchical) structure of the shell and the defence behaviour of the snails.

This concludes our answer to reviewer 1. We thank reviewer 1 for rising concerns.

Reviewer: 2

Comments to the Author(s)

This new version has been improved regard to the comments of the reviewers. It remains some informations that do not appear in this new text. Actually a paper must be understandable in its own form and not besides a previous paper : it does not take several lines to add some informations that strengthen the results. It is very important to understand that the experiment correspond to a “real” prey-predator situation to avoid bias. Especially, some informations that can be found in the ref 16 will gain to figure in this text.

- Origin of the shell: You answer to my question about the localities that all the locations have been indicated in ref 16 (Morii et al., 2016). But in the ref 16 there were more than 30 shells listed. And you have measured 6 shells of *K. editha*. Please indicate from where they are coming from.

In addition to the information that we had indicated in the Materials and Methods section (“The snails were collected in Bibai, Hokkaido, Japan (43.3285°N, 141.9688°E) in June 2010.”), we now also provide the list of location, collection dates and sample IDs in the supplementary materials: “The list of location and collection dates are referenced in the supplementary materials (Table S1).”

Table S1. Collection details of the snails.

Sample ID	Family	Species	Locality name	Latitude (N°)	Longitude (E°)	Altitude (m)	Collected date	Collector	Nbr of samples	Adult or juvenile
M3322	Camæniidae (Bradybaenidae)	Karaftohelix gainesi	Bibai, Hokkaido, Japan	43.32853	141.96875	190	24-Jun-2010	Yuta MORII	1	Adult
M3323	Camæniidae (Bradybaenidae)	Karaftohelix gainesi	Bibai, Hokkaido, Japan	43.32853	141.96875	190	24-Jun-2010	Yuta MORII	1	Adult
M3324	Camæniidae (Bradybaenidae)	Karaftohelix gainesi	Bibai, Hokkaido, Japan	43.32853	141.96875	190	24-Jun-2010	Yuta MORII	1	Adult
M3325	Camæniidae (Bradybaenidae)	Karaftohelix gainesi	Bibai, Hokkaido, Japan	43.32853	141.96875	190	24-Jun-2010	Yuta MORII	1	Adult
M3326	Camæniidae (Bradybaenidae)	Karaftohelix gainesi	Bibai, Hokkaido, Japan	43.32853	141.96875	190	24-Jun-2010	Yuta MORII	1	Adult
M3327	Camæniidae (Bradybaenidae)	Karaftohelix gainesi	Bibai, Hokkaido, Japan	43.32853	141.96875	190	24-Jun-2010	Yuta MORII	1	Adult
M3328	Camæniidae (Bradybaenidae)	Karaftohelix gainesi	Bibai, Hokkaido, Japan	43.32853	141.96875	190	24-Jun-2010	Yuta MORII	1	Adult
M3329	Camæniidae (Bradybaenidae)	Karaftohelix gainesi	Bibai, Hokkaido, Japan	43.32853	141.96875	190	24-Jun-2010	Yuta MORII	4	Adult
M3334	Camæniidae (Bradybaenidae)	Karaftohelix editha	Bibai, Hokkaido, Japan	43.32853	141.96875	190	24-Jun-2010	Yuta MORII	1	Adult
M3335	Camæniidae (Bradybaenidae)	Karaftohelix editha	Bibai, Hokkaido, Japan	43.32853	141.96875	190	24-Jun-2010	Yuta MORII	1	Adult
M3336	Camæniidae (Bradybaenidae)	Karaftohelix editha	Bibai, Hokkaido, Japan	43.32853	141.96875	190	24-Jun-2010	Yuta MORII	1	Adult
M3337	Camæniidae (Bradybaenidae)	Karaftohelix editha	Bibai, Hokkaido, Japan	43.32853	141.96875	190	24-Jun-2010	Yuta MORII	1	Adult
M3338	Camæniidae (Bradybaenidae)	Karaftohelix editha	Bibai, Hokkaido, Japan	43.32853	141.96875	190	24-Jun-2010	Yuta MORII	1	Adult
M3339	Camæniidae (Bradybaenidae)	Karaftohelix editha	Bibai, Hokkaido, Japan	43.32853	141.96875	190	24-Jun-2010	Yuta MORII	1	Adult

- The fact the both predators come from the same locality and the both prey species come from the same locality also must be indicated.

Several statements indicating that the 2 species are sharing the same habitat and most of their genetic content are present in the text.

For example, in the introduction: “In Hokkaido, two malacophagous carabid beetles, *Damaster blaptoides* and *Acoptolabrus gehinii* (Carabidae, Coleoptera) are dominant, and are likely the largest predators of *K. editha* and *K. gainesi* so far.”;

in section 3.1: “The two snail species studied share the same habitat, have closely related genomes [18], but exhibit two distinct macroscopic phenotypes.”,

and: “In this environment, they are facing similar predatory attacks mostly from carabid beetles *Damaster blaptoides* and *Acoptolabrus gehinii* [16,17].”

We added several other statements to further strengthen the case, such as:

in the introduction: “Among nine *Karaftohelix* species, genetically related and sharing the same habitat, two alternative anti-predator behaviours, passive and active, were observed.”

- It lacks information about the videos/movies analysed, please indicate the number of the movies found in ref 16 (Morii et al., 2016) or better the link.

We added the following to the Materials and Method section of the paper: “

The movies used to observe the response of the snails to predatory attacks from

beetles can be found at this link: <https://www.nature.com/articles/srep35600#Sec14>.

The movies 2, 6 and 9 were analysed for the swinging behaviour whereas the movie 8

was used to look at the retraction of the snail into its shell. The analysis of those

movies was done using iMovie (Apple Inc., USA) and Image J (NIH, USA).”

I agree with the referee 2 concerning the size of the figure 1 but for the right part B (very-too small). One observation concerning images of *K. gainesi*: the predator does not change its position during the rotation of the shell whereas in movie of the ref 16, the predator moves. Is there an adaption of the shell position regard to the position of the predator (not seen by the prey because the eyes are located at the opposite side)

We increased the size of Figure 1 part B. We did not observe an adaption of the shell position with regards to the position of the predator. Under attack of the beetle, it was found that the predator was approaching closely the snail first. Then, upon swinging its shell, the beetle was scared away and could not approach the snail closely anymore.

In the conclusion, the Figure 5 has not been more explained or justified by other data, actually the data inside are partial regard to the subject and do not allow the links (correlations?) shown (the mechanical strength, the insertion of muscles and many other things have to be studied). I suggest again to remove it or at least to remove the arrows. In a “table” format with simple columns, the different characteristics appear as a summary of the results and not as an attempt to establish relationships (the legend has to be changed).

We took into account the reviewer’s concerns and changed “inter-relationships” with “potential correlations” (lesser strong word) in the title of the manuscript, now: “Structure-behaviour correlations between two genetically closely related snail species”, in the text, and in the caption of the figure 5: “Correlations between the shell structure, mechanical properties, functions and the snail defence behaviour for *K. editha* and *K. gainesi*.”

We replaced “can” by “could” in the conclusion.

We also specified in the conclusions that it is indeed challenging to generalise the work since only 2 species were studied, and that the primary aim of the work was to compare those 2 species only. Indeed, it is quite rare to find 2 genetically closely related species living in the same environment, sharing the same habitat and predators, and exhibiting diverging defence behaviour. We are nevertheless convinced that the comparison of the 2 species in terms of shell structure, composition and the final performance and functions of the shells is meaningful to an interdisciplinary community as it illustrates how material design/structure impacts the final use and properties. This study highlights the implications in natural samples, but, from an engineering perspective, extrapolations could be drawn.

We added the following sentence to the conclusions: “Although it is challenging to draw general conclusions when comparing only two species, this study illustrates how two structures with similar ultimate goal can be designed in different ways to realize different specific functions.”

Minor comments

There is no more cone illustration in figure 1C. Change the text line 52 p6 (and line 56- p.8?) We corrected now.

P6 line 20-21: precise (if correct) that apparently (?) there is no preference or specificity between the two prey-predators.

We explain in the manuscript that the 2 preys are facing similar attacks from the beetles. We added the word “similar” to increase clarity: “In this environment, they are facing similar predatory attacks mostly from carabid beetles *Damaster blaptoides* and *Acoptolabrus gehinii* [16,17].”

P7 line 57: “similar shell development and formation”: at which level? What is the difference between development and formation? We rephrased with the following: “This could suggest that the shells of the two species followed similar formation path.”

Line 11 p2 & Line 17 p 13 Minus character for mantis shrimp. We corrected now.

Line 11 p13 these and not theses We corrected.

Line 24 p13 specimens and not specimen. We corrected now.

We thank the reviewer for the helpful comments and suggestions, and the time dedicated to our work.

Reviewer: 3

Comments to the Author(s)

The authors present interesting results from a comparison of two closely-related terrestrial gastropod species that exhibit different predator avoidance strategies. They found that *K. editha* that retreats into its shell when confronted with beetle predators has a harder more stable shell and that *K. gainensi* that swings its shell to club the predator has a more lightweight shell that allows for greater movement. These shell characteristics would seem to make sense with regard to their predator avoidance strategies. I do have a few suggestions, comments, and questions.

1. Page 2, line 18. The word "and" should be inserted between "patterns" and "chirality." We corrected now

2. Page 2, line 22. "divergence" should be "divergences." We corrected now

3. Page 2, line 59. Are the two species that show the behavioral response sister species or did this trait evolve more than once? Has a phylogeny of this genus been done? This would be interesting to know.

Thank you for this question, indeed the two species share most of their genetic material. Some phylogenetic information is provided in ref 16.

4. Page 4, line 9. I think a better wording would be "In total, 6 shells from living adult *Karafkaohelix editha* and 11 from living adult . . ." Then remove the words "alive" from the next sentence.

Thanks, we corrected with: "In total, 6 shells from living adult *Karafkaohelix editha* and 11 from living adult *Karafkaohelix gainesi* were studied."

Page 5, Figure 1D. It is not clear in this figure which set of points is which species. This should be defined in the figure heading.

We clarified on the figure which set of points corresponds to which species, as well as in the caption: (black, *K. editha* and white, *K. gainesi*) was added to Figure 1 and Figure 2.

Page 7. Do the angles described in figure 2 change during development? I wonder if the differences in strategy may in part be explained by the differences in adult size.

This is an interesting question. We had also collected juvenile of those species and they presented similar angles and shells. Investigation at early stages of development would be required but we are unfortunately unable to provide those data.

Page 8, line 23. I think an explanation of "shell moment" would be beneficial to many readers.

Thanks for this suggestion, we added the following sentence: "The shell moment describes the rotation of the shell around the columella, under an applied force: gravity when the snail is at rest, or the muscle force of the snail during swinging."

Page 8, line 49. The authors suggest that the snails evolved their shell morphology first and their defense strategy second. Could the reverse be true. Could the changes in shell be selected for due to difference in defense strategy?

We do not think that the snails had their shell evolve first, then their defence strategy. Indeed, the reverse could be true, as well as concomitant changes. To clarify this, we pointed this out in the conclusions: “Although we listed correlations between the structure of the shells and the behaviour of snail, we cannot predict from our analysis if the shell evolved first, then the behaviour, or the opposite, or if these changes occurred concomitantly. It is however legitimate to assume that each structure has evolved to be optimized for each defence scenario.”

Page 11, line 13.

I think a definition and explanation of the importance of Young's modulus is needed here. We added this explanation: “Nanoindentation is a method that provides the local Young's modulus of a material, i.e. its stiffness.”

Page 13, line 28. The authors use the term "significantly" to describe difference in shell thickness. The term significant in science is very much tied to statistics. No statistics were done to compare the thicknesses that I can see, so I suggest using a different word such as "much" or "substantially."

We removed “significantly” from the sentence.

Page 14, lines 12-31. The authors discuss the differences in the organization of the shell between the species. Two snails were chosen for analysis for each species. Was any variation found within species?

The images presented in the manuscript are representative images across the species. We did not observe much variation within species.

Page 15, Conclusion. My primary concern with this paper is that the authors show a correlation between predator avoidance strategy and morphology based on only two species. One could make a much stronger case if multiple species were examined representing each strategy. The authors state that there are two species that exhibit the behavioral response and multiple other species that retract into their shells when confronted with the predator. It is dangerous to make conclusions based on a sample size of 2. I realize that this paper represents a substantial amount of work. I suggest, at the very least, that the authors acknowledge the limitation and danger of basing conclusions on just two species.

We agree with the reviewer that it is difficult to generalize our study given the limited amount of species sharing same genes, same habitat and same predator but dissimilar defence behaviours. The primary goal of the work was, for this reason, only based on the comparison of those 2 snail species. We acknowledged this now in our conclusion with: “Although it is challenging to draw general conclusions when comparing only two species, this study illustrates how two structures with similar ultimate goal can be designed in different ways to realize different specific functions.” In addition, to nuance our statements, we changed “interrelationships” with “correlations” in the title of the manuscript, now: “Structure-behaviour correlations between two genetically closely related snail species”, in the text, and in the caption of the figure 5: “Correlations between the shell structure, mechanical properties, functions and the snail defence behaviour for *K. editha* and *K. gainesi*.” We also replace “can” by “could” in the conclusion.

This ends our response to Reviewer 3 and we thank him/her for the time dedicated to our manuscript and the overall positive appreciation of our work.